# Learning Influence Functions from Incomplete Observations

**Xinran He**     **Ke Xu**     **David Kempe**     **Yan Liu**
University of Southern California, Los Angeles, CA 90089
{xinranhe, xuk, dkempe, yanliu.cs}@usc.edu

## Abstract

We study the problem of learning influence functions under incomplete observations of node activations. Incomplete observations are a major concern as most (online and real-world) social networks are not fully observable. We establish both proper and improper PAC learnability of influence functions under randomly missing observations. Proper PAC learnability under the Discrete-Time Linear Threshold (DLT) and Discrete-Time Independent Cascade (DIC) models is established by reducing incomplete observations to complete observations in a modified graph. Our improper PAC learnability result applies for the DLT and DIC models as well as the Continuous-Time Independent Cascade (CIC) model. It is based on a parametrization in terms of reachability features, and also gives rise to an efficient and practical heuristic. Experiments on synthetic and real-world datasets demonstrate the ability of our method to compensate even for a fairly large fraction of missing observations.

## 1 Introduction

Many social phenomena, such as the spread of diseases, behaviors, technologies, or products, can naturally be modeled as the diffusion of a contagion across a network. Owing to the potentially high social or economic value of accelerating or inhibiting such diffusions, the goal of understanding the flow of information and predicting information cascades has been an active area of research [10, 7, 9, 14, 1, 20]. A key task here is learning *influence functions*, mapping sets of initial adopters to the individuals who will be influenced (also called *active*) by the end of the diffusion process [10].

Many methods have been developed to solve the influence function learning problem [9, 7, 5, 8, 3, 16, 18, 24, 25]. Most approaches are based on fitting the parameters of a diffusion model based on observations, e.g., [8, 7, 18, 9, 16]. Recently, Du et al. [3] proposed a *model-free* approach to learn influence functions as coverage functions; Narasimhan et al. [16] establish proper PAC learnability of influence functions under several widely-used diffusion models.

All existing approaches rely on the assumption that the observations in the training dataset are complete, in the sense that all active nodes are observed as being active. However, this assumption fails to hold in virtually all practical applications [15, 6, 2, 21]. For example, social media data are usually collected through crawlers or acquired with public APIs provided by social media platforms, such as Twitter or Facebook. Due to non-technical reasons and established restrictions on the APIs, it is often impossible to obtain a complete set of observations even for a short period of time. In turn, the existence of unobserved nodes, links, or activations may lead to a significant misestimation of the diffusion model's parameters [19, 15].

We take a step towards addressing the problem of learning influence functions from incomplete observations. Missing data are a complicated phenomenon, but to address it meaningfully and rigorously, one must make at least *some* assumptions about the process resulting in the loss of data. We focus on *random* loss of observations: for each activated node independently, the node's activation is observed only with probability $r$, the *retention rate*, and fails to be observed with probability

$1 - r$. Random observation loss naturally occurs when crawling data from social media, where rate restrictions are likely to affect all observations equally.

We establish both proper and improper PAC learnability of influence functions under incomplete observations for two popular diffusion models: the Discrete-Time Independent Cascade (DIC) and Discrete-Time Linear Threshold (DLT) models. In fact, randomly missing observations do not even significantly increase the required sample complexity. The result is proved by interpreting the incomplete observations as complete observations in a transformed graph,

The PAC learnability result implies good sample complexity bounds for the DIC and DLT models. However, the PAC learnability result does not lead to an *efficient* algorithm, as it involves marginalizing a large number of hidden variables (one for each node not observed to be active).

Towards designing more practical algorithms and obtaining learnability under a broader class of diffusion models, we pursue improper learning approaches. Concretely, we use the parameterization of Du et al. [3] in terms of reachability basis functions, and optimize a modified loss function suggested by Natarajan et al. [17] to address incomplete observations. We prove that the algorithm ensures improper PAC learning for the DIC, DLT and Continuous-Time Independent Cascade (CIC) models. Experimental results on synthetic cascades generated from these diffusion models and real-world cascades in the MemeTracker dataset demonstrate the effectiveness of our approach. Our algorithm achieves nearly a 20% reduction in estimation error compared to the best baseline methods on the MemeTracker dataset.

Several recent works also aim to address the issue of missing observations in social network analysis, but with different emphases. For example, Chierichetti et al. [2] and Sadikov et al. [21] mainly focus on recovering the *size* of a diffusion process, while our task is to learn the influence functions from several incomplete cascades. Myers et al. [15] mainly aim to model unobserved external influence in diffusion. Duong et al. [6] examine learning diffusion models with missing links from *complete* observations, while we learn influence functions from incomplete cascades with missing activations. Most related to our work are papers by Wu et al. [23] and simultaneous work by Lokhov [13]. Both study the problem of network inference under incomplete observations. Lokhov proposes a dynamic message passing approach to marginalize all the missing activations, in order to infer diffusion model parameters using maximum likelihood estimation, while Wu et al. develop an EM algorithm. Notice that the goal of learning the model parameters differs from our goal of learning the influence functions directly. Both [13] and [23] provide empirical evaluation, but do not provide theoretical guarantees.

## 2 Preliminaries

### 2.1 Models of Diffusion and Incomplete Observations

**Diffusion Model.** We model propagation of opinions, products, or behaviors as a diffusion process over a social network. The social network is represented as a directed graph $G = (V, E)$, where $n = |V|$ is the number of nodes, and $m = |E|$ is the number of edges. Each edge $e = (u, v)$ is associated with a parameter $w_{uv}$ representing the strength of influence user $u$ has on $v$. We assume that the graph structure (the edge set $E$) is known, while the parameters $w_{uv}$ are to be learned. Depending on the diffusion model, there are different ways to represent the strength of influence between individuals. Nodes can be in one of two states, *inactive* or *active*. We say that a node gets activated if it adopts the opinion/product/behavior under the diffusion process. In this work, we focus on *progressive* diffusion models, where a node remains active once it gets activated.

The diffusion process begins with a set of seed nodes (initial adopters) $S$, who start active. It then proceeds in discrete or continuous time: according to a probabilistic process, additional nodes may become active based on the influence from their neighbors. Let $N(v)$ be the in-neighbors of node $v$ and $A_t$ the set of nodes activated by time $t$. We consider the following three diffusion models:

**Discrete-time Linear Threshold (DLT) model [10]:** Each node $v$ has a threshold $\theta_v$ drawn independently and uniformly from the interval $[0, 1]$. The diffusion under the DLT model unfolds in discrete time steps: a node $v$ becomes active at step $t$ if the total incoming weight from its active neighbors exceeds its threshold: $\sum_{u \in N(v) \cap A_{t-1}} w_{uv} \geq \theta_v$.

**Discrete-time Independent Cascade (DIC) model [10]:** The DIC model is also a discrete-time model. The weight $w_{uv} \in [0, 1]$ captures an activation probability. When a node $u$ becomes active in step $t$, it attempts to activate all currently inactive neighbors in step $t + 1$. For each neighbor $v$, it

succeeds with probability $w_{uv}$. If it succeeds, $v$ becomes active; otherwise, $v$ remains inactive. Once $u$ has made all these attempts, it does not get to make further activation attempts at later times.

**Continuous-time Independent Cascade (CIC) model [8]:** The CIC model unfolds in continuous time. Each edge $e = (u, v)$ is associated with a delay distribution with $w_{uv}$ as its parameter. When a node $u$ becomes newly active at time $t$, for every neighbor $v$ that is still inactive, a delay time $d_{uv}$ is drawn from the delay distribution. $d_{uv}$ is the duration it takes $u$ to activate $v$, which could be infinite (if $u$ does not succeed in activating $v$). Nodes are considered activated by the process if they are activated within a specified observation window $[0, \tau]$.

Fix one of the diffusion models defined above and its parameters. For each seed set $S$, let $\Delta_S$ be the distribution of final active sets. (In the case of the DIC and DLT model, this is the set of active nodes when no new activations occur; for the CIC model, it is the set of nodes active at time $\tau$.) For any node $v$, let $F_v(S) = \mathrm{Prob}_{A \sim \Delta_S}[v \in A]$ be the (marginal) probability that $v$ is activated according to the dynamics of the diffusion model with initial seeds $S$. Then, define the *influence function* $\boldsymbol{F} : 2^V \to [0, 1]^n$ mapping seed sets to the vector of marginal activation probabilities: $\boldsymbol{F}(S) = [F_1(S), \dots, F_n(S)]$. Notice that the marginal probabilities do not capture the full information about the diffusion process contained in $\Delta_S$ (since they do not observe co-activation patterns), but they are sufficient for many applications, such as influence maximization [10] and influence estimation [4].

**Cascades and Incomplete Observations.** We focus on the problem of learning influence functions from cascades. A cascade $C = (S, A)$ is a realization of the random diffusion process; $S$ is the set of seeds and $A \sim \Delta_S, A \supseteq S$ is the set of activated nodes at the end of the random process. Similar to Narasimhan et al. [16], we focus on *activation-only* observations, namely, we only observe *which nodes* were activated, but not *when* these activations occurred.[1]

To capture the fact that some of the node activations may have been unobserved, we use the following model of independently randomly missing data: for each (activated) node $v \in A \setminus S$, the activation of $v$ is actually *observed* independently with probability $r$. With probability $1 - r$, the node's activation is unobservable. For seed nodes $v \in S$, the activation is never lost. Formally, define $\tilde{A}$ as follows: each $v \in S$ is deterministically in $\tilde{A}$, and each $v \in A \setminus S$ is in $\tilde{A}$ independently with probability $r$. Then, the incomplete cascade is denoted by $\tilde{C} = (S, \tilde{A})$.

## 2.2 Objective Functions and Learning Goals
To measure estimation error, we primarily use a quadratic loss function, as in [16, 3]. For two $n$-dimensional vectors $\boldsymbol{x}, \boldsymbol{y}$, the quadratic loss is defined as $\ell_{\mathrm{sq}}(\boldsymbol{x}, \boldsymbol{y}) = \frac{1}{n} \cdot ||\boldsymbol{x} - \boldsymbol{y}||_2^2$. We also use this notation when one or both arguments are sets: when an argument is a set $S$, we formally mean to use the *indicator function* $\chi_S$ as a vector, where $\chi_S(v) = 1$ if $v \in S$, and $\chi_S(v) = 0$ otherwise. In particular, for an activated set $A$, we write $\ell_{\mathrm{sq}}(A, \boldsymbol{F}(S)) = \frac{1}{n}||\chi_A - \boldsymbol{F}(S)||_2^2$.

We now formally define the problem of learning influence functions from incomplete observations. Let $\mathcal{P}$ be a distribution over seed sets (i.e., a distribution over $2^V$), and fix a diffusion model $\mathcal{M}$ and parameters, together giving rise to a distribution $\Delta_S$ for each seed set. The algorithm is given a set of $M$ incomplete cascades $\tilde{\mathcal{C}} = \{(S_1, \tilde{A}_1), \dots, (S_M, \tilde{A}_M)\}$, where each $S_i$ is drawn independently from $\mathcal{P}$, and $\tilde{A}_i$ is obtained by the incomplete observation process described above from the (random) activated set $A_i \sim \Delta_{S_i}$. The goal is to learn an influence function $\boldsymbol{F}$ that accurately captures the diffusion process. Accuracy of the learned influence function is measured in terms of the squared error with respect to the true model: $\mathrm{err}_{\mathrm{sq}}[\boldsymbol{F}] = \mathbb{E}_{S \sim \mathcal{P}, A \sim \Delta_S} [\ell_{\mathrm{sq}}(A, \boldsymbol{F}(S))]$. That is, the expectation is over the seed set and the randomness in the diffusion process, but not the data loss process.

**PAC Learnability of Influence Functions.** We characterize the learnability of influence functions under incomplete observations using the Probably Approximately Correct (PAC) learning framework [22]. Let $\mathcal{F}_{\mathcal{M}}$ be the class of influence functions under the diffusion model $\mathcal{M}$, and $\mathcal{F}_{\mathcal{L}}$ the class of influence functions the learning algorithm is allowed to choose from. We say that $\mathcal{F}_{\mathcal{M}}$ is PAC learnable if there exists an algorithm $\mathcal{A}$ with the following property: for all $\varepsilon, \delta \in (0, 1)$, all parametrizations of the diffusion model, and all distributions $\mathcal{P}$ over seed sets $S$: when given *activation-only*

and *incomplete* training cascades $\tilde{\mathcal{C}} = \{(S_1, \tilde{A}_1), \ldots, (S_M, \tilde{A}_M)\}$ with $M \geq poly(n, m, 1/\varepsilon, 1/\delta)$, $\mathcal{A}$ outputs an influence function $\boldsymbol{F} \in \mathcal{F}_{\mathcal{L}}$ satisfying $\mathrm{Prob}_{\tilde{\mathcal{C}}}[\mathrm{err}_{\mathrm{sq}}[\boldsymbol{F}] - \mathrm{err}_{\mathrm{sq}}[\boldsymbol{F}^*] \geq \varepsilon] \leq \delta$.

Here, $\boldsymbol{F}^* \in \mathcal{F}_{\mathcal{M}}$ is the ground truth influence function. The probability is over the training cascades, including the seed set generation, the stochastic diffusion process, and the missing data process. We say that an influence function learning algorithm $\mathcal{A}$ is *proper* if $\mathcal{F}_{\mathcal{L}} \subseteq \mathcal{F}_{\mathcal{M}}$; that is, the learned influence function is guaranteed to be an instance of the true diffusion model. Otherwise, we say that $\mathcal{A}$ is an *improper* learning algorithm.

## 3 Proper PAC Learning under Incomplete Observations

In this section, we establish proper PAC learnability of influence functions under the DIC and DLT models. For both diffusion models, $\mathcal{F}_{\mathcal{M}}$ can be parameterized by an edge parameter vector $\boldsymbol{w}$, whose entries $w_e$ are the activation probabilities (DIC model) or edge weights (DLT model). Our goal is to find an influence function $\boldsymbol{F}^{\boldsymbol{w}} \in \mathcal{F}_{\mathcal{M}}$ that outputs accurate marginal activation probabilities. While our goal is *proper* learning — meaning that the function must be from $\mathcal{F}_{\mathcal{M}}$ — we do *not* require that the inferred parameters match the true edge parameters $\boldsymbol{w}$. Our main theoretical results are summarized in Theorems 1 and 2.

**Theorem 1.** *Let $\lambda \in (0, 0.5)$. The class of influence functions under the DIC model in which all edge activation probabilities satisfy $w_e \in [\lambda, 1 - \lambda]$ is PAC learnable under incomplete observations with retention rate $r$. The sample complexity[2] is $\tilde{O}(\frac{n^3 m}{\varepsilon^2 r^4})$.*

**Theorem 2.** *Let $\lambda \in (0, 0.5)$, and consider the class of influence functions under the DLT model such that the edge weight for every edge satisfies $w_e \in [\lambda, 1 - \lambda]$, and for every node $v$, $1 - \sum_{u \in N(v)} w_{uv} \in [\lambda, 1 - \lambda]$. This class is PAC learnable under incomplete observations with retention rate $r$. The sample complexity is $\tilde{O}(\frac{n^3 m}{\varepsilon^2 r^4})$.*

In this section, we present the intuition and a proof sketch for the two theorems. Details of the proof are provided in Appendix B.

The key idea of the proof of both theorems is that a set of incomplete cascades $\tilde{\mathcal{C}}$ on $G$ under the two models can be considered as essentially complete cascades on a transformed graph $\hat{G} = (\hat{V}, \hat{E})$. The influence functions of nodes in $\hat{G}$ can be learned using a modification of the result of Narasimhan et al. [16]. Subsequently, the influence functions for $G$ are directly obtained from the influence functions for $\hat{G}$, by exploiting that influence functions only focus on the marginal activation probabilities.

The transformed graph $\hat{G}$ is built by adding a layer of $n$ nodes to the graph $G$. For each node $v \in V$ of the original graph, we add a new node $v' \in V'$ and a directed edge $(v, v')$ with known and fixed edge parameter $\hat{w}_{vv'} = r$. (The same parameter value serves as activation probability under the DIC model and as edge weight under the DLT model.) The new nodes $V'$ have no other incident edges, and we retain all edges $e = (u, v) \in E$. Inferring their parameters is the learning task.

For each observed (incomplete) cascade $(S_i, \tilde{A}_i)$ on $G$ (with $S_i \subseteq \tilde{A}_i$), we produce an observed activation set $A'_i$ as follows: (1) for each $v \in \tilde{A}_i \setminus S_i$, we let $v' \in A'_i$ deterministically; (2) for each $v \in S_i$ independently, we include $v' \in A'_i$ with probability $r$. This defines the training cascades $\hat{\mathcal{C}} = \{(S_i, A'_i)\}$.

Now consider any edge parameters $\boldsymbol{w}$, applied to both $G$ and the first layer of $\hat{G}$. Let $\boldsymbol{F}(S)$ denote the influence function on $G$, and $\hat{\boldsymbol{F}}(S) = [\hat{F}_{1'}(S), \ldots, \hat{F}_{n'}(S)]$ the influence function of the nodes in the added layer $V'$ of $\hat{G}$. Then, by the transformation, for all nodes $v \in V$, we get that

$$\hat{F}_{v'}(S) = r \cdot F_v(S). \tag{1}$$

And by the definition of the observation loss process, $\mathrm{Prob}[v \in \tilde{A}_i] = r \cdot F_v(S) = \hat{F}_{v'}(S)$ for all non-seed nodes $v \notin S_i$.

While the cascades $\hat{\mathcal{C}}$ are not complete on all of $\hat{G}$, in a precise sense, they provide complete information on the activation of nodes in $V'$. In Appendix B, we show that Theorem 2 of Narasimhan et al. [16] can be extended to provide identical guarantees for learning $\hat{\boldsymbol{F}}(S)$ from the modified

observed cascades $\hat{\mathcal{C}}$. For the DIC model, this is a straightforward modification of the proof from [16]. For the DLT model, [16] had not established PAC learnability[3], so we provide a separate proof.

Because the results of [16] and our generalizations ensure *proper* learning, they provide edge weights $\boldsymbol{w}$ between the nodes of $V$. We use these exact same edge weights to define the learned influence functions in $G$. Equation (1) then implies that the learned influence functions on $V$ satisfy $F_v(S) = \frac{1}{r} \cdot \hat{F}_{v'}(S)$. The detailed analysis in Appendix B shows that the learning error only scales by a multiplicative factor $\frac{1}{r^2}$.

The PAC learnability result shows that there is no information-theoretical obstacle to influence function learning under incomplete observations. However, it does not imply an *efficient* algorithm. The reason is that a hidden variable would be associated with each node not observed to be active, and computing the objective function for empirical risk minimization would require marginalizing over all of the hidden variables. The proper PAC learnability result also does not readily generalize to the CIC model and other diffusion models, even under complete observations. This is due to the lack of a succinct characterization of influence functions as under the DIC and DLT models. Therefore, in the next section, we explore improper learning approaches with the goal of designing practical algorithms and establishing learnability under a broader class of diffusion models.

## 4 Efficient Improper Learning Algorithm

Instead of parameterizing the influence functions using the edge parameters, we adopt the model-free influence function learning framework, InfluLearner, proposed by Du et al. [3] to represent the influence function as a sum of weighted basis functions. From now on, we focus on the influence function $F_v(S)$ of a single fixed node $v$.

**Influence Function Parameterization.** For all three diffusion models (CIC, DIC and DLT), the diffusion process can be characterized equivalently using live-edge graphs. Concretely, the results of [10, 4] state that for each instance of the CIC, DIC, and DLT models, there exists a distribution $\Gamma$ over live-edge graphs $H$ assigning probability $\gamma_H$ to each live-edge graph $H$ such that $F_v^*(S) = \sum_{H:\text{at least one node in } S \text{ has a path to } v \text{ in } H} \gamma_H$.

To reduce the representation complexity, notice that from the perspective of activating $v$, two different live-edge graphs $H, H'$ are "equivalent" if $v$ is reachable from exactly the same nodes in $H$ and $H'$. Therefore, for any node set $T$, let $\beta_T^* := \sum_{H:\text{exactly the nodes in } T \text{ have paths to } v \text{ in } H} \gamma_H$. We then use characteristic vectors as feature vectors $\boldsymbol{r}_T = \boldsymbol{\chi}_T$, where we will interpret the entry for node $u$ as $u$ having a path to $v$ in a live-edge graph. More precisely, let $\phi(x) = \min\{x, 1\}$, and $\boldsymbol{\chi}_S$ the characteristic vector of the seed set $S$. Then, $\phi(\boldsymbol{\chi}_S^\top \cdot \boldsymbol{r}_T) = 1$ if and only if $v$ is reachable from $S$, and we can write $F_v^*(S) = \sum_T \beta_T^* \cdot \phi(\boldsymbol{\chi}_S^\top \cdot \boldsymbol{r}_T)$.

This representation still has exponentially many features (one for each $T$). In order to make the learning problem tractable, we sample a smaller set $\mathcal{T}$ of $K$ features from a suitably chosen distribution, implicitly setting the weights $\beta_T$ of all other features to 0. Thus, we will parametrize the learned influence function as $F_v^{\boldsymbol{\beta}}(S) = \sum_{T \in \mathcal{T}} \beta_T \cdot \phi(\boldsymbol{\chi}_S^\top \cdot \boldsymbol{r}_T)$.

The goal is then to learn weights $\beta_T$ for the sampled features. (They will form a distribution, i.e., $||\boldsymbol{\beta}||_1 = 1$ and $\boldsymbol{\beta} \geq 0$.) The crux of the analysis is to show that a sufficiently small number $K$ of features (i.e., sampled sets) suffices for a good approximation, and that the weights can be learned efficiently from a limited number of observed incomplete cascades. Specifically, we consider the log likelihood function $\ell(t, y) = y \log t + (1 - y) \log(1 - t)$, and learn the parameter vector (a distribution) by maximizing the likelihood $\sum_{i=1}^M \ell(F_v^{\boldsymbol{\beta}}(S_i), \chi_{A_i}(v))$.

**Handling Incomplete Observations.** The maximum likelihood estimation cannot be directly applied to incomplete cascades, as we do not have access to $A_i$ (only the incomplete version $\tilde{A}_i$). To address this issue, notice that the MLE problem is actually a binary classification problem with log loss and $y_i = \chi_{A_i}(v)$ as the label. From this perspective, incompleteness is simply class-conditional noise on the labels. Let $\tilde{y}_i = \chi_{\tilde{A}_i}(v)$ be our *observation* of whether $v$ was activated or not under the incomplete cascade $i$. Then, $\text{Prob}[\tilde{y}_i = 1 | y_i = 1] = r$ and $\text{Prob}[\tilde{y}_i = 1 | y_i = 0] = 0$. In words,

the incomplete observation $\tilde{y}_i$ suffers from one-sided error compared to the complete observation $y_i$. By results of Natarajan et al. [17], we can construct an unbiased estimator of $\ell(t, y)$ using only the incomplete observations $\tilde{y}$, as in the following lemma.

**Lemma 3** (Corollary of Lemma 1 of [17]). *Let $y$ be the true activation of node $v$ and $\tilde{y}$ the incomplete observation. Then, defining $\tilde{\ell}(t, y) := \frac{1}{r} y \log t + \frac{2r-1}{r}(1-y)\log(1-t)$, for any $t$, we have $\mathbb{E}_{\tilde{y}}\left[\tilde{\ell}(t, \tilde{y})\right] = \ell(t, y)$.*

Based on this lemma, we arrive at the final algorithm of solving the maximum likelihood estimation problem with the adjusted likelihood function $\tilde{\ell}(t, y)$:

$$\begin{aligned}\text{Maximize} \quad & \sum_{i=1}^{M} \tilde{\ell}(F_v^{\boldsymbol{\beta}}(S_i), \chi_{\tilde{A}_i}(v)) \\ \text{subject to} \quad & ||\boldsymbol{\beta}||_1 = 1, \boldsymbol{\beta} \geq 0.\end{aligned} \tag{2}$$

We analyze conditions under which the solution to (2) provides improper PAC learnability under incomplete observations; these conditions will apply for all three diffusion models.

These conditions are similar to those of Lemma 1 in the work of Du et al. [3], and concern the approximability of the reachability distribution $\beta_T^*$. Specifically, let $q$ be a distribution over node sets $T$ such that $q(T) \leq C\beta_T^*$ for all node sets $T$. Here, $C$ is a (possibly very large) number that we will want to bound below. Let $T_1, \ldots, T_K$ be $K$ i.i.d. samples drawn from the distribution $q$. The features are then $\boldsymbol{r}_k = \boldsymbol{\chi}_{T_k}$. We use the truncated version of the function $F_v^{\boldsymbol{\beta},\lambda}(S)$ with parameter[4] $\lambda$ as in [3]: $F_v^{\boldsymbol{\beta},\lambda}(S) = (1 - 2\lambda)F_v^{\boldsymbol{\beta}}(S) + \lambda$.

Let $\mathcal{M}_\lambda$ be the class of all such truncated influence functions, and $F_v^{\tilde{\boldsymbol{\beta}},\lambda} \in \mathcal{M}_\lambda$ the influence functions obtained from the optimization problem (2). The following theorem (proved in Appendix C) establishes the accuracy of the learned functions.

**Theorem 4.** *Assume that the learning algorithm uses $K = \tilde{\Omega}(\frac{C^2}{\varepsilon^2})$ features in the influence function it constructs, and observes[5] $M = \tilde{\Omega}(\frac{\log C}{\varepsilon^4 r^2})$ incomplete cascades with retention rate $r$. Then, with probability at least $1 - \delta$, the learned influence functions $F_v^{\tilde{\boldsymbol{\beta}},\lambda}$ for each node $v$ and seed distribution $\mathcal{P}$ satisfy $\mathbb{E}_{S\sim\mathcal{P}}\left[(F_v^{\tilde{\boldsymbol{\beta}},\lambda}(S) - F_v^*(S))^2\right] \leq \varepsilon$.*

The theorem implies that with enough incomplete cascades, an algorithm can approximate the ground truth influence function to arbitrary accuracy. Therefore, all three diffusion models are improperly PAC learnable under incomplete observations. The final sample complexity does not contain the graph size, but is implicitly captured by $C$, which will depend on the graph and how well the distribution $\beta_T^*$ can be approximated. Notice that with $r = 1$, our bound on $M$ has logarithmic dependency on $C$ instead of polynomial, as in [3]. The reason for this improvement is discussed further in Appendix C.

**Efficient Implementation.** As mentioned above, the features $T$ cannot be sampled from the exact reachability distribution $\beta_T^*$, because it is inaccessible (and complex). In order to obtain useful guarantees from Theorem 4, we follow the approach of Du et al. [3], and approximate the distribution $\beta_T^*$ with the product of the marginal distributions, estimated from observed cascades.

The optimization problem (2) is convex and can therefore be solved in time polynomial in the number of features $K$. However, the guarantees in Theorem 4 require a possibly large number of features. In order to obtain an *efficient* algorithm for practical use and our experiments, we sacrifice the guarantee and use a fixed number of features.

## 5 Experiments

In this section, we experimentally evaluate the algorithm from Section 4. Since no other methods explicitly account for incomplete observations, we compare it to several state-of-the-art methods for influence function learning with full information. Hence, the main goal of the comparison is to examine to what extent the impact of missing data can be mitigated by being aware of it. We compare

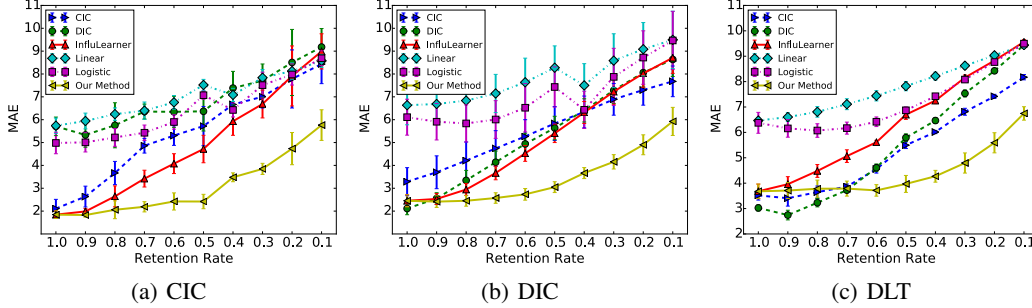

|            |            |            |
| :--------: | :--------: | :--------: |
| (a) CIC    | (b) DIC    | (c) DLT    |

Figure 1: MAE of estimated influence as a function of the retention rate on synthetic datasets for (a) CIC model, (b) DIC model, (c) DLT model. The error bars show one standard deviation.

our algorithm to the following approaches: (1) **CIC** fits the parameters of a CIC model, using the NetRate algorithm [7] with exponential delay distribution. (2) **DIC** fits the activation probabilities of a DIC model using the method in [18]. (3) **InfluLearner** is the model-free approach proposed by Du et al. in [3] and discussed in Section 4. (4) **Logistic** uses logistic regression to learn the influence functions $F_u(S) = f(\boldsymbol{\chi}_S^\top \cdot \boldsymbol{c}_u + b)$ for each $u$ independently, where $\boldsymbol{c}_u$ is a learnable weight vector and $f(x) = \frac{1}{1+e^{-x}}$ is the logistic function. (5) **Linear** uses linear regression to learn the total influence $\sigma(S) = \boldsymbol{c}^\top \cdot \boldsymbol{\chi}_S + b$ of the set $S$. Notice that the CIC and DIC methods have access to the *activation time* of each node in addition to the final activation status, giving them an inherent advantage.

## 5.1 Synthetic cascades

**Data generation.** We generate synthetic networks with core-peripheral structure following the Kronecker graph model [12] with parameter matrix $[0.9, 0.5; 0.5, 0.3]$.[6] Each generated network has $512$ nodes and $1024$ edges. Subsequently, we generate $8192$ cascades as training data using the CIC, DIC and DLT models, with random seed sets whose sizes are power law distributed. The retention rate is varied between $0.1$ and $0.9$. The test set contains $200$ independently sampled seed sets generated in the same way as the training data. Details of the data generation process are provided in Appendix A.

**Algorithm settings.** We apply all algorithms to cascades generated from all three models; that is, we also consider the results under model misspecification. Whenever applicable, we set the hyperparameters of the five comparison algorithms to the ground truth values. When applying the NetRate algorithm to discrete-time cascades, we set the observation window to $10.0$. When applying the method in [18] to continuous-time cascades, we round activation times up to the nearest multiple of $0.1$, resulting in $10$ discrete time steps. For the model-free approaches (InfluLearner and our algorithm), we use $K = 200$ features.

**Results.** Figure 1 shows the Mean Absolute Error (MAE) between the estimated total influence $\sigma(S)$ and the true influence value, averaged over all testing seed sets. For each setting (diffusion model and retention rate), the reported MAE is averaged over five independent runs.

The main insight is that accounting for missing observations indeed strongly mitigates their effect: notice that for retention rates as small as $0.5$, our algorithm can almost completely compensate for the data loss, whereas both the model-free and parameter fitting approaches deteriorate significantly even for retention rates close to 1. For the parameter fitting approaches, even such large retention rates can lead to missing and spurious edges in the inferred networks, and thus significant estimation errors. Additional observations include that fitting influence using (linear or logistic) regression does not perform well at all, and that the CIC inference approach appears more robust to model misspecification than the DIC approach.

**Sensitivity of retention rate.** We presented the algorithms as knowing $r$. Since $r$ itself is inferred from noisy data, it may be somewhat misestimated. Figure 2 shows the impact of misestimating $r$. We generate synthetic cascades from all three diffusion models with a true retention rate of $0.8$, and

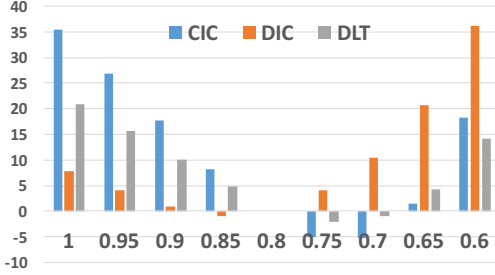

Figure 2: Relative error in MAE under retention rate misspecification. $x$-axis: retention rate $r$ used by the algorithm. $y$-axis: relative difference of MAE compared to using the true retention rate 0.8.

Figure 3: MAE of influence estimation on seven sets of real-world cascades with 20% of activations missing.

then apply our algorithm with (incorrect) retention rate $r \in \{0.6, 0.65, \ldots, 0.95, 1\}$. The results are averaged over five independent runs. While the performance decreases as the misestimation gets worse (after all, with $r = 1$, the algorithm is basically the same as InfluLearner), the degradation is graceful.

## 5.2 Influence Estimation on real cascades

We further evaluate the performance of our method on the real-world MemeTracker[7] dataset [11]. The dataset consists of the propagation of short textual phrases, referred to as *Memes*, via the publication of blog posts and main-stream media news articles between March 2011 and February 2012. Specifically, the dataset contains seven groups of cascades corresponding to the propagation of Memes with certain keywords, namely "apple and jobs", "tsunami earthquake", "william kate marriage'", "occupy wall-street", "airstrikes", "egypt" and "elections." Each cascade group consists of 1000 nodes, with a number of cascades varying from 1000 to 44000. We follow exactly the same evaluation method as Du et al. [3] with a training/test set split of 60%/40%.

To test the performance of influence function learning under incomplete observations, we randomly delete 20% of the occurrences, setting $r = 0.8$. The results for other retention rates are similar and omitted. Figure 3 shows the MAE of our methods and the five baselines, averaged over 100 random draws of test seed sets, for all groups of memes. While some baselines perform very poorly, even compared to the best baseline (InfluLearner), our algorithm provides an 18% reduction in MAE (averaged over the seven groups), showing the potential of data loss awareness to mitigate its effects.

## 6 Extensions and Future Work

In the full version available on arXiv, we show both experimentally and theoretically how to generalize our results to non-uniform (but independent) loss of node activations, and how to deal with a misestimation of the retention rate $r$. Any non-trivial partial information about $r$ leads to positive PAC learnability results.

A much more significant departure for future work would be dependent loss of activations, e.g., losing all activations of some randomly chosen nodes. As another direction, it would be worthwhile to generalize the PAC learnability results to other diffusion models, and to design an efficient algorithm with PAC learning guarantees.

## Acknowledgments

We would like to thank anonymous reviewers for useful feedback. The research was sponsored in part by NSF research grant IIS-1254206 and by the U.S. Defense Advanced Research Projects Agency (DARPA) under the Social Media in Strategic Communication (SMISC) program, Agreement Number W911NF-12-1-0034. The views and conclusions are those of the authors and should not be interpreted as representing the official policies of the funding agency or the U.S. Government.

## Footnotes

[1]Narasimhan et al. [16] refer to this model as *partial observations*; we change the terminology to avoid confusion with "incomplete observations."

[2]The $\tilde{O}$ notation suppresses poly-logarithmic dependence on $1/\lambda$, $1/\delta$, $n$, and $m$.

[3][16] shows that the DLT model with *fixed* thresholds is PAC learnable under complete cascades. We study the DLT model when the thresholds are uniformly distributed random variables.

[4] The proof of Theorem 4 in Appendix C will show how to choose $\lambda$.

[5] The $\tilde{\Omega}$ notation suppresses all logarithmic terms except $\log C$, as $C$ could be exponential or worse in the number of nodes.

[6]We also experimented on Kronecker graphs with hierarchical community structure ($[0.9, 0.1; 0.1, 0.9]$) and random structure ($[0.5, 0.5; 0.5, 0.5]$). The results are similar and omitted due to space constraints.

[7]We use the preprocessed version of the dataset released by Du et al. [3] and available at `http://www.cc.gatech.edu/~ndu8/InfluLearner.html`. Notice that the dataset is semi-real, as multi-node seed cascades are artificially created by merging single-node seed cascades.

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
