[Supplementary Material · incompleteFuncLearn_supp.pdf]

# A  Details on Synthetic Data Generation

Here, we provide details on the generation of synthetic cascades. The cascades are generated according to the CIC, DIC and DLT models. For the CIC model, we use an exponential delay distribution on each edge whose parameters are drawn independently and uniformly from $[0, 1]$. The observation window length is $\tau = 1.0$. For the DIC model, the activation probability for each edge is chosen independently and uniformly from $[0, 0.4]$. For the DLT model, we follow [10] and set the edge weight $w_{uv}$ as $1/d_v$ where $d_v$ is the in-degree of node $v$. For each model, we generate 8192 cascades as training data. The seed sets are sampled uniformly at random with sizes drawn from a power law distribution with parameter 2.5. The generated cascades have average sizes of 10.8, 12.8 and 13.0 in the CIC, DIC and DLT models, respectively. We then create incomplete cascades by varying the retention rate between 0.1 and 0.9. To sidestep the computational cost of running Monte Carlo simulations, we estimate the ground truth influence of the test seed sets using the method proposed in [3], with the true model parameters.

# B  Proofs for Section 3

## B.1  Proof of Theorem 1

Here, we flesh out the proof sketch from Section 3 for the DIC model. For the transformed graph $\hat{G}$, we consider only the influence functions of the $n$ nodes in the added layer $V'$. Recall that we write $\hat{F}(S) = [\hat{F}_{1'}(S), \ldots, \hat{F}_{n'}(S)]$ for the influence function of those nodes. Let $\hat{F}^*$ be the ground truth influence function for the same nodes, and $F^*$ the ground truth influence function for $G$. Let $\mathcal{M}(G)$ and $\mathcal{M}(\hat{G})$ be the class of influence functions of $G$ and $\hat{G}$. For functions $\hat{F}$, we write $\widehat{\text{err}}_{sq}[\hat{F}] = \mathbb{E}_{S,A}\left[\frac{1}{n}\sum_{v' \in V'}(\chi_A(v') - \hat{F}_{v'}(S))^2\right]$. Notice that the ground truth functions minimize the expected squared error, i.e., $\hat{F}^* \in \operatorname{argmin}_{\hat{F} \in \mathcal{M}(\hat{G})}\widehat{\text{err}}_{sq}[\hat{F}]$ and $F^* \in \operatorname{argmin}_{F \in \mathcal{M}(G)}\text{err}_{sq}[F]$. We will show that $\text{err}_{sq}[F] - \text{err}_{sq}[F^*]$ can be made arbitrary small.

We first prove a variation of Theorem 2 from [16] for learning $\hat{F}$, by verifying that all the supporting lemmas still apply. The modified Theorem 2 from [16] is the following:

**Theorem 5.** *Assume that the learning algorithm observes $M = \tilde{\Omega}(\hat{\epsilon}^{-2}n^3m)$ training cascades $\hat{\mathcal{C}} = \{(S_i, A_i')\}$ under the DIC model. Then, with probability at least $1 - \delta$, we have*

$$\widehat{err}_{sq}[\hat{F}] - \widehat{err}_{sq}[\hat{F}^*] \leq \hat{\epsilon}. \tag{3}$$

*Proof.* While the cascades in $\hat{\mathcal{C}}$ are incomplete on $V$, they are *complete* on $V'$. We use this completeness of the cascades as follows. Consider the restricted class of the DIC model on the transformed graph $\hat{G}$ in which only the $m$ activation probabilities $w$ between nodes in $V$ are learnable, while the edges $(v, v')$ have a fixed weight of $r$. Define the log-likelihood for a cascade $(S, A')$ as

$$\mathcal{L}(S, A'|w) = \sum_{v' \in V'} \chi_{A_i'}(v')\log(\hat{F}_{v'}^w(S)) + (1 - \chi_{A_i'}(v'))\log(1 - \hat{F}_{v'}^w(S)).$$

The algorithm outputs an influence function $\hat{F}$ based on the solution of the following optimization problem:

$$w^* \in \operatorname{argmax}_{w \in [\lambda, 1-\lambda]^m} \sum_{i=1}^M \mathcal{L}(S_i, A_i'|w).$$

As the function $\hat{F}$ is learned from the DIC model, Lemma 3 in [16] carries thorough to establish the Lipschitz continuity of DIC influence functions.

**Lemma 6** (Lipschitz continuity of DIC influence). *Fix $S \subseteq V$ and $v' \in V'$. For any $w, w' \in \mathbb{R}^m$ with $||w - w'||_1 \leq \epsilon$, we have $|\hat{F}_{v'}^w(S) - \hat{F}_{v'}^{w'}(S)| \leq \epsilon$.*

Moreover, such instances (on $2n$ nodes) still only have $m$ parameters, and the $L_\infty$ covering number bound in Lemma 8 from [16] applies without any changes.

**Lemma 7** (Covering number of DIC influence functions). *The $L_\infty$ covering number of the restricted class of the DIC influence functions on the transformed graph for radius $\epsilon$ is $O((m/\epsilon)^m)$.*

Establishing the sample complexity bound on the log-likelihood objective (Lemma 4 in [16]) requires that all function values be bounded away from 0 and 1 (Lemma 9 in [16]). We assume that $r < 1$, as Lemma 4 in [16] directly holds when there are no missing data at all. Let $\lambda > 0$ be the bound on the edge activation probabilities in $G$ from our Theorem 1; that is, $\lambda \le w_{uv} \le 1 - \lambda$ for all $u, v \in V$. Due to the layered structure of $\hat{G}$, we have that $r \cdot \lambda^n \le \hat{F}_{v'}(S) \le r \cdot (1 - \lambda^n)$.[8] Therefore, Lemma 4 in [16] carries thorough with the same sample complexity of $\tilde{O}(\hat{\epsilon}^{-2} n^3 m)$:

**Lemma 8** (Sample complexity guarantee on the log-likelihood objective). *Fix $\epsilon, \delta \in (0, 1)$ and $M = \Omega(n^2 \ln(1/\lambda)^2 \frac{m \ln(m/\hat{\epsilon}) + nm \ln(1/\lambda) + \ln(1/\delta)}{\hat{\epsilon}^2}) = \tilde{\Omega}(\hat{\epsilon}^{-2} n^3 m)$. With probability at least $1 - \delta$ (over the draws of the training cascades),*

$$\max_{\boldsymbol{w} \in [\lambda, 1-\lambda]^m} \mathbb{E}_{S,A'} \left[ \frac{1}{n} \mathcal{L}(S, A'|\boldsymbol{w}) \right] - \mathbb{E}_{S,A'} \left[ \frac{1}{n} \mathcal{L}(S, A'|\boldsymbol{w}^*) \right] \le \epsilon.$$

As all the lemmas used in the proof of Theorem 2 from [16] remain true, we have proved our Theorem 5, with the guarantee that $\mathrm{err}_{\mathrm{sq}}[\hat{\boldsymbol{F}}] - \mathrm{err}_{\mathrm{sq}}[\hat{\boldsymbol{F}}^*] \le \hat{\epsilon}$. $\qquad \square$

Finally, we recall that according to Equation (1), $F_v(S) = \frac{1}{r} \cdot \hat{F}_{v'}(S)$ and $F_v^*(S) = \frac{1}{r} \cdot \hat{F}_{v'}^*(S)$, giving us that

$$
\begin{aligned}
\mathrm{err}_{\mathrm{sq}}[\boldsymbol{F}] - \mathrm{err}_{\mathrm{sq}}[\boldsymbol{F}^*] \quad &\overset{(*)}{=} \quad \frac{1}{n} \sum_{v \in V} \mathbb{E}_S \left[ (F_v(S) - F_v^*(S))^2 \right] \\
&\overset{\text{Equation (1)}}{=} \quad \frac{1}{n} \sum_{v' \in \hat{V}} \mathbb{E}_S \left[ (\frac{1}{r} \hat{F}_{v'}(S) - \frac{1}{r} \hat{F}_{v'}^*(S))^2 \right] \\
&\overset{(*)}{=} \quad \frac{\widehat{\mathrm{err}}_{\mathrm{sq}}[\hat{\boldsymbol{F}}] - \widehat{\mathrm{err}}_{\mathrm{sq}}[\hat{\boldsymbol{F}}^*]}{r^2} \\
&\overset{\text{Equation (3)}}{\le} \quad \frac{\hat{\epsilon}}{r^2}
\end{aligned}
$$

(The steps labeled (*) are applications of Equation (4) from [16].) Now, by taking $\hat{\epsilon} = \varepsilon \cdot r^2$, with $\tilde{O}(\frac{n^3 m}{\varepsilon^2 r^4})$ incomplete cascades, we obtain that $\mathrm{err}_{\mathrm{sq}}[\boldsymbol{F}] - \mathrm{err}_{\mathrm{sq}}[\boldsymbol{F}^*] \le \varepsilon$.

### B.2 Proof of Theorem 2

We will show that the analogue of Theorem 5 for the DLT model also holds. We do so by following the same sequence of steps as in Appendix B.1 and verifying that all the steps in the proof of Theorem 2 in [16] still hold. The main difference is that a new proof is needed for establishing Lipschitz continuity of the DLT influence function with respect to the $L_1$ norm (the analogue of Lemma 3 in [16]). We begin by establishing this lemma.

**Lemma 9** (Lipschitz continuity). *Fix $S \subseteq V$ and $u \in V$. For any $\boldsymbol{w}, \boldsymbol{w}' \in \mathbb{R}^m$ with $||\boldsymbol{w} - \boldsymbol{w}'||_1 \le \varepsilon$, we have that $|F_u^{\boldsymbol{w}}(S) - F_u^{\boldsymbol{w}'}(S)| \le \varepsilon$.*

*Proof.* As shown in [10], the influence functions under the DLT model can be also characterized via the reachability under a distribution over live-edge graphs. Specifically, the distribution is as follows [10, Theorem 2.5]: for each node $v$, pick at most one of its incoming edges at random, selecting the edge from $z \in N(v)$ with probability $w_{zv}$ and selecting no incoming edge with probability $1 - \sum_{z \in N(v)} w_{zv}$. For each node $v$, let the random variable $X_v$ be the incoming neighbor chosen for $v$, with $X_v = \perp$ if $v$ has no incoming edge. For simplicity of notation, we define $w_{\perp v} = 1 - \sum_{z \in N(v)} w_{zv}$. Define $\boldsymbol{X} = (X_v)_{v \in V}$, and write $\mathcal{X}$ for the set of all such vectors $\boldsymbol{X}$. For

any node $v$, we write $\mathcal{X}_{-v}$ for the set of all vectors with edges (or $\perp$) for all nodes except $v$. And for a vector $\boldsymbol{X} \in \mathcal{X}_{-v}$, we write $\boldsymbol{X}[v \mapsto z]$ for the vector in which all entries agree with those in $\boldsymbol{X}$, except for the entry for $v$ which is now $z$.

Let $R_{\boldsymbol{X}}(v, S)$ be the indicator function of whether node $v$ is reachable from the seed set $S$ in the graph $(V, \boldsymbol{X})$, where we interpret $\boldsymbol{X}$ as the set of all edges $(X_v, v)$ with $X_v \neq \perp$. Claim 2.6 of [10] implies that

$$F_u^{\boldsymbol{w}}(S) = \sum_{\boldsymbol{X}} \prod_{v \in V} w_{X_v v} R_{\boldsymbol{X}}(u, S).$$

We fix an edge $(y, y')$ and take the partial derivative of $F_u^{\boldsymbol{w}}(S)$ with respect to $w_{yy'}$:

$$
\begin{aligned}
\left| \frac{\partial F_u^{\boldsymbol{w}}(S)}{\partial w_{yy'}} \right| &= \left| \frac{\partial}{\partial w_{yy'}} \left( \sum_{z \in N(y) \cup \{\perp\}} w_{zy} \sum_{\boldsymbol{X} \in \mathcal{X}_{-y}} \prod_{v \in V \setminus \{y\}} w_{X_v v} \cdot R_{\boldsymbol{X}[y \mapsto z]}(u, S) \right) \right| \\
&= \left| \sum_{\boldsymbol{X} \in \mathcal{X}_{-y}} \prod_{v \in V \setminus \{y\}} w_{X_v v} \cdot R_{\boldsymbol{X}[y \mapsto y']}(u, S) - \sum_{\boldsymbol{X} \in \mathcal{X}_{-y}} \prod_{v \in V \setminus \{y\}} w_{X_v v} \cdot R_{\boldsymbol{X}[y \mapsto \perp]}(u, S) \right| \\
&\leq \left| \sum_{\boldsymbol{X} \in \mathcal{X}_{-y}} \prod_{v \in V \setminus \{y\}} w_{X_v v} \right| \\
&= 1.
\end{aligned}
$$

Therefore, $\|\nabla_{\boldsymbol{w}} F_u^{\boldsymbol{w}}(S)\|_\infty \leq 1$, implying Lipschitz continuity. $\qquad \square$

Next, we bound the values of the influence functions away from 0 and 1. Because each edge weight $w_{zv} \in [\lambda, 1-\lambda]$ by assumption, and we further assumed that $w_{\perp v} = 1 - \sum_{z \in N(v)} w_{zv} \in [\lambda, 1-\lambda]$, it follows directly (as in the proof for the DIC model) that $r \cdot \lambda^n \leq \hat{F}_{v'}(S) \leq r \cdot (1 - \lambda^n)$. This establishes the analogue of Lemma 9 in [16], and we can therefore apply Lemma 4 in [16], obtaining a sample complexity of $\tilde{O}(\hat{\epsilon}^{-2} n^3 m)$ under the DLT model. As all the used lemmas remain true, the results of Theorem 5 also hold for the DLT model. Finally, exploiting the same relation between $\boldsymbol{F}(S)$ and $\hat{\boldsymbol{F}}(S)$ as in the proof of Theorem 1 leads to the conclusion of Theorem 2.

## C Proof of Theorem 4

Let $M = \tilde{\Omega}(\frac{\log C}{\epsilon^4 r^2})$, and let $F_v^{\tilde{\boldsymbol{\beta}}, \lambda}(S)$ be the influence functions obtained in Theorem 4. We will show that for any single node $v$, with probability at least $1 - \delta/n$,

$$\mathbb{E}_S \left[ (F_v^{\tilde{\boldsymbol{\beta}}, \lambda}(S) - F_v^*(S))^2 \right] \leq \varepsilon.$$

The theorem then follows by taking a union bound over all $n$ nodes.

Recall that $\mathcal{M}_\lambda$ is the function class of all truncated influence functions. We write

$$\mathcal{R}_M(\mathcal{M}_\lambda) := \mathbb{E}_{S_i \sim \mathcal{P}, (\epsilon_i)_i \sim \text{Uniform}(\{-1,1\}^M)} \left[ \sup_{\boldsymbol{F} \in \mathcal{M}_\lambda} \frac{1}{M} \sum_{i=1}^M \epsilon_i \cdot F_v(S_i) \right]$$

for its Rademacher complexity, where the $\epsilon_i$'s are i.i.d. Rademacher (symmetric Bernoulli) random variables. By Lemma 12 in [3], there exists a truncated influence function $F_v^{\hat{\boldsymbol{\beta}}, \lambda} \in \mathcal{M}_\lambda$ with $K = O(\frac{C^2}{\varepsilon^2} \log \frac{Cn}{\varepsilon \delta})$ features such that $\mathbb{E}_{S \sim \mathcal{P}} \left[ (F_v^{\hat{\boldsymbol{\beta}}, \lambda}(S) - F_v^*(S))^2 \right] \leq 2\varepsilon^2 + 2\lambda^2$ with probability at least $1 - \frac{\delta}{2n}$.

Using the log likelihood function $\ell(t, y) = y \log t + (1-y) \log(1-t)$ as defined in Section 4, we write the log loss of the influence function $F_v$ as $\text{err}_{\log}[F_v] = \mathbb{E}_{S, A} [-\ell(F_v(S), A)]$. By Lemma 2 in [17], with probability at least $1 - \frac{\delta}{2n}$,

$$\text{err}_{\log}[F_v^{\tilde{\boldsymbol{\beta}}, \lambda}] \leq \min_{f \in \mathcal{M}_\lambda} \text{err}_{\log}[f] + \frac{4}{\lambda \cdot r} \mathcal{R}_M(\mathcal{M}_\lambda) + \sqrt{\frac{\log(2n/\delta)}{2M}}.$$

Because $F_v^{\hat{\beta},\lambda} \in \mathcal{M}_\lambda$, we can bound that $\min_{f \in \mathcal{M}_\lambda} \mathrm{err}_{\log}[f] \leq \mathrm{err}_{\log}[F_v^{\hat{\beta},\lambda}(S)]$ on the right-hand side. Subtracting $\mathrm{err}_{\log}[F_v^*]$ from both sides, we obtain

$$\mathrm{err}_{\log}[F_v^{\tilde{\beta},\lambda}] - \mathrm{err}_{\log}[F_v^*] \quad \leq \quad \mathrm{err}_{\log}[F_v^{\hat{\beta},\lambda}] - \mathrm{err}_{\log}[F_v^*] + \frac{4}{\lambda \cdot r}\mathcal{R}_M(\mathcal{M}_\lambda) + \sqrt{\frac{\log(2n/\delta)}{2M}}. \quad (4)$$

The square and log errors can be related to each other as in the proof of Theorem 2 in [16], as follows:

$$\mathbb{E}_S\left[(F_v^{\tilde{\beta},\lambda}(S) - F_v^*(S))^2\right] \quad \leq \quad \frac{1}{2}(\mathrm{err}_{\log}[F_v^{\tilde{\beta},\lambda}] - \mathrm{err}_{\log}[F_v^*]).$$

Hence, in order to obtain a bound on $\mathbb{E}_S\left[(F_v^{\tilde{\beta},\lambda}(S) - F_v^*(S))^2\right]$, it suffices to upper-bound the right-hand side of (4). The term $\mathrm{err}_{\log}[F_v^{\hat{\beta},\lambda}] - \mathrm{err}_{\log}[F_v^*]$ can be bounded as in the proof of Lemma 2 in [3], using Lemma 11 and Lemma 16 from [3]: Assume that $F_v^{\hat{\beta},\lambda}$ uses $K = \Omega(\frac{C^2}{\hat{\epsilon}^2} \log \frac{Cn}{\hat{\epsilon}\hat{\delta}})$ features. Then, with probability at least $1 - \hat{\delta}$, we have

$$\mathrm{err}_{\log}[F_v^{\hat{\beta},\lambda}] - \mathrm{err}_{\log}[F_v^*] \quad \leq \quad \frac{\hat{\epsilon}^2 + \lambda^2}{\lambda}(1 + \log\frac{1}{\lambda}). \quad (5)$$

Next, we bound the Rademacher complexity of the function class $\mathcal{M}_\lambda$:

**Lemma 10.** *The Rademacher complexity $\mathcal{R}_M(\mathcal{M}_\lambda)$ for the function class $\mathcal{M}_\lambda$ with at most $K$ features is at most $\sqrt{\frac{2\log(1+K)}{M}}$.*

*Proof.* Recall that we use basis functions $\phi_i(S) := \min\{1, \chi_S^\top r_{T_i}\}$. Let $\mathcal{W} = \{\phi_i | i = 1, \ldots, K\} \cup \{\mathbb{1}\}$, where $\mathbb{1}$ is the constant function with value 1. By definition, we have $\mathcal{M}_\lambda \subseteq \mathrm{conv}(\mathcal{W})$, where $\mathrm{conv}(\mathcal{W})$ denotes the convex hull. Therefore, $\mathcal{R}_M(\mathcal{M}_\lambda) \leq \mathcal{R}_M(\mathrm{conv}(\mathcal{W})) = \mathcal{R}_M(\mathcal{W})$. Since $|\phi_i(S)| \leq 1$, by Massart's finite lemma[9], we have $\mathcal{R}_M(\mathcal{W}) \leq \sqrt{\frac{2\log(1+K)}{M}}$, completing the proof. $\square$

To finish the proof of Theorem 4, let $\epsilon$ be the desired accuracy. Define $\hat{\delta} = \frac{\delta}{2n}$ and $\hat{\epsilon} = \lambda = \frac{\epsilon}{c' \log \frac{1}{\epsilon}}$, where $c'$ is a sufficiently large constant. Then, the right-hand side of (5) is upper-bounded by $\hat{\epsilon} \cdot (1 + \log\frac{1}{\hat{\epsilon}}) \leq \frac{\epsilon}{2}$.

With $M = \Omega(\frac{\log K}{\epsilon^4 r^2})$, we have $\frac{4}{\lambda \cdot r}\mathcal{R}_M(\mathcal{M}_\lambda) \leq \frac{\epsilon}{4}$. Whenever $M = \Omega(\frac{\log\frac{n}{\delta}}{\epsilon^2})$, we also get that $\sqrt{\frac{\log(n/\delta)}{2M}} \leq \frac{\epsilon}{4}$. Taking $M$ as the maximum of the above three, which is satisfied when $M = \tilde{\Omega}(\frac{\log C}{\epsilon^4 r^2})$, we can substitute all of the bounds into the right-hand side of (4) and obtain that $\mathbb{E}_S\left[(F_v^{\tilde{\beta},\lambda}(S) - F_v^*(S))^2\right] \leq \epsilon$ with probability at least $1 - \frac{\delta}{n}$. Now, taking a union bound over all nodes $v$ concludes the proof.

**Discussion.** Notice that when the retention rate is 1, our Theorem 4 significantly improves the sample complexity bound compared to Du et al. [3]. The sample complexity in [3] is $\tilde{O}(\frac{C^2}{\epsilon^3})$, while our theorem implies a sample complexity of $\tilde{O}(\frac{\log C}{\epsilon^4})$ under complete observations. The improvement is derived from bounding the Rademacher complexity of the function class $\mathcal{M}_\lambda$ instead of the $L_{2,\infty}$ dimension. The Rademacher bound leads to a logarithmic dependence of the sample complexity on the number of features $K$, whereas the $L_{2,\infty}$ bound results in a polynomial dependence.

## Footnotes

[8]As in the proof of Lemma 4 in [16], we assume that there exists a path in the graph $\hat{G}$ from a node in $S$ to node $v'$; the cases where this assumption fails can be handled easily.

[9]Massart's finite lemma states the following: Let $\mathcal{F}$ be a finite set of functions, such that $\sup_{f \in \mathcal{F}} \frac{1}{n}\sum_{i=1}^n f(X_i)^2 \leq C^2$ for any variables values $X_1, \ldots, X_n$. Then, the Rademacher complexity of $\mathcal{F}$ is upper bounded by $\mathcal{R}_n(\mathcal{F}) \leq \sqrt{\frac{2C^2 \log |\mathcal{F}|}{n}}$.