[Reviews · NeurIPS 2016]

Reviewer 1

Summary

The authors consider the problem of PAC-learning the influence function in cascade models under a incomplete observation model. For each observed cascade, only the initial seeds and random subset of final active nodes are observed---specifically, each active node is observed as being active independently with probability 1-mu. The specific learning goal is to estimate the marginal probabilities of each node being activated given any source set S. The paper considers both proper PAC learning (the estimated influence function corresponds to an actual diffusion model) and improper PAC learning (the estimated influence function is parameterized differently). For proper learning, they extend results of [14] to the case of incomplete observations. The main idea is to reduce (by a fairly straightforward reduction) the problem of learning with incomplete cascades to the problem of learning with complete observations in a modified graph (and with modified training cascades). They show that influence functions are PAC-learnable with a sample complexity that increases by a factor of 1/(1-mu)^4 over that of learning with complete cascades. However, proper PAC learning is computationally intractable, so they instead consider improper learning. For improper learning, they extend the results of [2] for "model-free" learning of influence functions to the case of incomplete observations. In particular, the influence function is estimated by approximating the reverse reachable sets of each node v as a mixture of sets sampled from some distribution q that approximates the true one. The results are parameterized in terms of how good an approximation q is. The sample complexity increases by a factor of 1/(1-mu)^2 compared with learning with complete cascades. They also improve the analysis of [2] to show a better dependence (logarithmic instead of polynomial) on the parameter C that quantifies how good an approximation q is. Experiments compare with a number of different baselines to show that their model-free method performs much better with an increasing fraction of missing observations.

Qualitative Assessment

In general, the paper is extremely well written and technically sound. In terms of significance and impact, I appreciate the general line of work of directly estimating influence functions from pairs of seed and activation sets. This seems a lot more practically applicable than related problems of estimating true parameters or recovering the edges of an unobserved graph because it can be applied directly in a predictive setting. The specific high-level problem tackled by this paper---of estimating influence functions when some fraction of observations are missing---is relevant but not obviously a problem of huge impact. However, the problem is addressed very nicely. Two previous results are extended to the new observation model (refs [2,14]), the latter using another relatively recent result about learning with noisy labels (ref [15]). Each of the two major results in this paper required careful adaptation (and not just reuse) of the previous results. The experiments were thorough and showed improvements compared with a number of baselines. A limitation of the paper are the assumptions that: (1) the missing probability mu is known and, (2) observations are missing at random. While this is a reasonable place to start, the paper would be stronger if the authors acknowledge and discuss these limitations (e.g. lines 39-41 argue in favor of the assumption but do not acknowledge limitations). I also suggest that the tradeoffs in Section 4 be clarified. It is unclear leading up to Theorem 4 where the distribution q for generating features comes from, and whether the final choice (lines 264-265: "approximate \beta_T^* with the product of marginal distributions") carries any theoretical guarantee. The footnote suggests that C may be exponential in the number of nodes, but Theorem 4 asserts that on the order of C^2 features are required for the theoretical guarantee. How big is C in practice? Is it known? A better discussion of these issues is warranted. Minor comments: I believe these two references are also relevant. [1] estimates diffusion parameters with missing data. [2] recovers the edges of a graph from activation-only cascades. [1] Parameter Learning for Latent Network Diffusion. Wu, Kumar, Sheldon, Zilberstein. IJCAI 2013. [2] Learning form contagion (without timestamps). Amin, Heidari, Kearns, ICML 2014. In Section 2.1, clarify whether G and {w_{uv}} are known or unknown. In Theorems 1 and 2, it is somewhat confusing that the dependence of the sample complexity on delta is suppressed by the \tilde{O} notation. From the previous discussion, I expected to see delta in these expressions. The assertion in Lines 180-181 seems redundant with the previous equation. Isn't this case (non-seed nodes) already included in the statement of Eq. (1)? Line 195: I suggest saying the "computing the objective function" and not "writing down the objective function" is hard. For the experiments, one other baseline to consider is a naive adjustment to the predictions of the parametric models (CIC, DIC, etc.) that multiplies the predicted influence by 1/(1-mu). This would correct for missingness in the prediction phase but not the training phase. Was this already considered?

Confidence in this Review

2-Confident (read it all; understood it all reasonably well)


Reviewer 2

Summary

This paper address the problem of learning an influence function for the diffusion of an activation process on a graph where the information is incomplete. This problem has real world applications in many areas including the spread of disease or technology adoption. The influence estimation function has been studied but not in the context of missing informations which refers to some node-activations being unknown. The authors consider several diffusion models and both proper and improper learning. The final algorithm solves a maximum likelihood problem. Although there is not a lot of related work in the area, the authors compare to several algorithms that were not designed for incomplete data, and find that the proposed approach has the lowest mean absolute error.

Qualitative Assessment

his paper was very well written, was clear, and the technicality was high. Further the authors address an interesting and novel problem, and the experimentation and comparison with previous work is thorough. The authors consider several common diffusion models and use the PAC learning framework. My main concern regarding the paper is the practicality of the approach. While the authors make an effort to improve the efficiency of the approach, the algorithm relies on sampling many infections on the same graph. This does not seem to be realistic in many scenarios since a given infection occurs once on the graph. For example, researchers do not have the luxury of measring a disease spreading on a given population several times. In the real world dataset the authors experiments with, the authors treat every meme as a separate diffusion within the topic hence each topic has over 1000 samples, but this seems unlikely in other settings. Further, is each of the 1000 nodes activated at some point? Or are there nodes that are never activated? Along these lines, how are the results affected by the number of seed-sets/diffusions that are available? If a data-analysts has x-number of diffusion, can they know if this is enough for a good estimation? Another experiment that would be meaningful to see is the running time comparison with existing methods. For example, for a loss rate of 0.2, the DlC and InfluLeaner are competitive, how do they compare in running time? Why does the error increase for low loss rates? Overall the work is novel and influential, but I think is could benefit from a better explanation of the practicality of the approach in practice. As the authors said the random model for missing information will probably not hold in practice, but this work is a good first step for the problem.

Confidence in this Review

2-Confident (read it all; understood it all reasonably well)


Reviewer 3

Summary

The paper analyzes the sample complexity of learning influence functions for some well known models, in a setting where observations are missing at random. The authors give learnability proofs for a proper learning and for an improper hypothesis class previously suggested. The paper includes some experimental results on synthetic and semi-real data.

Qualitative Assessment

The paper is well written and presented in a clear manner (except perhaps the section on the reachability hypothesis class). The problem of missing observations is both important and challenging, and the positive theoretical results are encouraging and in place. The main concern however is novelty. The learnability proofs are, as the authors frankly mentioned themselves, mostly a variation on those found in [14]. The extension to DLT is an important contribution, but not a major. Other points are: 1) The notion of uniformly random missing observations is not very well motivated (but is, understandably, probably easier to analyze). 2) It is unclear why the stated objective is over individual nodes - most literature, as well as the experiments in the paper itself, asses the estimation of overall influence of a seed, which is a more reasonable goal. Defining a loss over individual nodes seems to be an overkill, and could be the reason for the large terms in the sample complexity bounds (e.g. thm. 4). 3) In thm. 1 - what is the dependence on lambda? Why does it have to be >0? At least in DIC, having lambda=0 is perfectly reasonable, and probably prevalent. 4) The method of [2], on which the improper suggestion is based, relies heavily on finding a distribution q which is close to beta*. As there are no guarantees on the compatibility of a marginal distribution, and since K scales as C^2 (which is generally unknown), drawing meaningful conclusions is not straightforward. This is probably the main weakness of the suggested method. 5) The literature contains other improper methods for influence estimation, e.g. 'Discriminative Learning of Infection Models' [WSDM 16], which can probably be modified to handle noisy observations. 6) The authors discuss the misestimation of mu, but as it is the proportion of missing observations - it is not wholly clear how it can be estimated at all. 5) The experimental setup borrowed from [2] is only semi-real, as multi-node seed cascades are artificially created by merging single-node seed cascades. This should be mentioned clearly. 7) As noted, the assumption of random missing entries is not very realistic. It would seem worthwhile to run an experiment to see how this assumption effects performance when the data is missing due to more realistic mechanisms.

Confidence in this Review

3-Expert (read the paper in detail, know the area, quite certain of my opinion)


Reviewer 4

Summary

This paper extends on recent work in learning graphs/influence functions in graphs by considering the case where observations are missing uniformly at random. This is a natural next step for this literature, and the authors extend previous work on PAC-learnability. They further extend previous work by Du et al. to provide a polynomial-time algorithm, relaxing the guarantee by limiting the number of required features.

Qualitative Assessment

Overall, the paper takes a solid step forward at answering the question of the learnability of graphs from incomplete observations: the theoretical results are extended in a constructive way. The generalization of the coverage function approximation by Du et al. to this setting is also interesting. The authors show that the suggested algorithm improves on prior work empirically. Error bars should be added to figure 3 which discusses the most interesting setting of real-world cascade. It is difficult to get a sense of the significance of the improvement over InfluLearner. Finally, the setting of information missing uniformly at random is a good first step, but ultimately very restricted. The paper could be strengthened by suggesting other specific models of missing data mechanisms, and how to tackle them.

Confidence in this Review

2-Confident (read it all; understood it all reasonably well)


Reviewer 5

Summary

This paper discusses the PAC learnability of influence functions over multiple cascades. Theoretical results are established for discrete time independent cascade and discrete time linear threshold models. The results are then extended to the case in which the influence statuses of some nodes are missing.

Qualitative Assessment

I'm not particularly familiar with the influence estimation or PAC learnability literature, but the paper seems to be well-written and sufficiently interesting. The results seem to be nice, as finding ways to handle missing data is important. I'm not sure influence is something with incredibly broad appeal to those at NIPS, which makes somewhat limits the usefulness. But there are a reasonable number of people in cs/ee/stat who are interested in problems of influence; so there should be a place for this at NIPS, barring substantial errors.

Confidence in this Review

2-Confident (read it all; understood it all reasonably well)


Reviewer 6

Summary

The paper provides sample complexities of proper PAC learning of the influence function in discrete-time independent cascade model and linear threshold model. Then, it gives the sample complexity of improper learning of the influence function in continuous-time independent cascade model and proposes a learning method to address the partial observation issue by adjusting the objective function of reference 2. Empirical results show that only given partial cascades, the algorithm achieves better performance than other baselines.

Qualitative Assessment

The paper discusses PAC learnability of influence functions under different diffusion models given partial observations of cascades, which provides interesting extensions in theory to existing work. However, the proof techniques used for the major theorems are mostly adopted and modified from reference 14 and 3 so as to adapt to the partial observation setting. Furthermore, the parametrization of the learning algorithm is the same as in reference 3. A direct application of reference 15 leads to the modification of the objective function in reference 3 to consider the missing probability. Essentially, the novelty of the current paper in theory is decreased by taking these factors into consideration. In addition, although given partial cascade observations, existing baselines will perform worse compared to the proposed algorithm. However, in practice, the missing probability is never known, and the proposed algorithm cannot learn it from the data either. This actually restricts the flexibility and applicability of the proposed algorithm further.

Confidence in this Review

3-Expert (read the paper in detail, know the area, quite certain of my opinion)